# Potential Effects of Climate and Human Influence Changes on Range and Diversity of Nine Fabaceae Species and Implications for Nature's Contribution to People in Kenya

**Risper Nyairo *** **and Takashi Machimura**

Graduate School of Engineering, Osaka University, 2-1, Yamadaoka, Suita Osaka 565-0871, Japan; mach@see.eng.osaka-u.ac.jp
* Correspondence: risper@ge.see.eng.osaka-u.ac.jp

**Abstract:** Climate and land-use changes are the main drivers of species distribution. On the basis of current and future climate and socioeconomic scenarios, species range projections were made for nine species in the Fabaceae family. Modeled species have instrumental and relational values termed as nature's contribution to people (NCP). For each species, five scenarios were analyzed resulting in 45 species range maps. Representative concentration pathway (RCP) 4.5 and three shared socioeconomic pathways (SSPs 1, 2, and 3) were used in the analysis. Species ranges under these scenarios were modeled using MaxEnt; a niche modeling software that relates species occurrence with environmental variables. Results were used to compute species richness and evenness based on Shannon's diversity Index. Results revealed a mix of range expansion and contraction for the modeled species. The findings highlighted which species may remain competitive in an urbanized future and which ones are detrimentally affected by climate. Parts of the country where species abundances are likely to change due to climate and socioeconomic changes were identified. Management of species will be required in people-dominated landscapes to maintain interactions between nature and society, while avoiding natural resource degradation and loss of NCP.

**Keywords:** climate change; human influence index (HII); Kenya; MaxEnt; NCP; species distribution; SSP scenarios

## 1. Introduction

Human wellbeing and the environment are the main focuses of sustainability [1]. These two entities are interdependent, that is, natural resources have many use (instrumental) and non-use (relational) values recently described as nature's contributions to people (NCP) by Diaz et al. [2], whereas humans are managers of these resources. In order to bring out the reciprocal obligations between people and nature, NCP is assessed in general as well as context-specific cultural perspectives [3], in which the role of indigenous and local knowledge is recognized. Maintenance of NCP is important for keeping human options open and enhancing quality of life [4]. NCP can either be positively or negatively affected by socioeconomic and demographic factors [2]. The UN Millenium Assessment, in 2005, established that demographic change was the major driver of land cover change. In turn, land cover change, along with climate change [5], are the direct drivers of biodiversity change. Changing climate may alter the establishment of species and lead to loss of NCP [6]. Globally, species are already responding to climatic changes [7,8].

Africa is thought to be the most vulnerable continent to both climate and land-use changes. The Intergovernmental Science-Policy Platform on Biodiversity and Ecosystem Services (IPBES) [9]

report on the status of biodiversity in Africa enumerated the synergistic influence of climate and anthropogenic change on biodiversity. According to the report, rapid population growth in the region, leading to urbanization, is among key factors affecting biodiversity. It is projected that the African population will double to 2.5 billion by 2050 and that half of the population will live in cities [10]. It is also estimated that climate change could result in the loss of about 5000 African plant species [11]. Therefore, it is important to map and monitor species responses to ongoing change for effective management. This need is especially great in Kenya where rapid demographic changes are occurring, but climate impact studies still remain scanty.

Current projections reveal a high rate of urbanization across Kenya driven by population growth [12] and rural-urban migration. The population of Kenya doubled between the years 1981 and 2003 [13] and close to 30% of the Kenyan population has now been classified as urban. According to KC and Lutz [14], projections of the Kenyan population estimate as high as 78 million people in 2050 and 96 million people by 2100 if current trends continue. Moreover, the shores of Lake Victoria have been projected to have one of the highest increases in urban land cover by 2030 [15]. In addition to population pressure, the impacts of climate change on plants have also been reported [16]. This is proof that natural landscapes and systems will undergo inevitable change. One way of determining where changes may occur is through the use of scenarios and models. Scenarios describe possible futures for drivers of change, while models translate those scenarios into projected consequences for nature and NCP.

Ecological niche modeling [17] is the process of relating species presence data with environmental variables. Among niche models, maximum entropy (MaxEnt) is popular because it can handle complex relationships, performs well with small datasets [18], and is able to rely on only presence data [19]. To make up for the lack of absence data, MaxEnt uses randomly drawn background samples. These pseudo-absences represent differences in climate, socioeconomic, topographic, and edaphic factors. By using MaxEnt, both bioclimatic and socioeconomic predictors can be combined to determine species niches. The model also allows prediction of future species distributions by swapping current predictors with their future projections. The shared socioeconomic pathways (SSPs) are new scenarios defining the trajectory of our world in terms of socioeconomic conditions in this century. In the SSP framework, there are five different storylines that include trends in land use [20]. Main differences among SSPs stem from assumptions on population growth, access to education, urbanization, economic growth, technology, among others. The SSPs were integrated in the Coupled Model Intercomparison Project (CMIP6) that utilized the new set of emission scenarios known as representative concentration pathways (RCPs) [21] to give climate change projections. The combination of SSP and RCP scenarios in modeling studies is paramount to contribute to sustainable development.

In this study, scenarios were applied to assess the influence of climate and socioeconomic change on potential distribution and diversity of species in the Fabaceae family, one of the plant families documented in Kenya [22]. The family contains the acacia trees that dot the landscape and have been exploited for several decades by local communities for products such as gum [23], firewood, construction materials [24], fodder [25], honey, cover cropping, hedging, and medicinal purposes [26,27]. Although species in the Fabaceae family have instrumental and relational values, there is insufficient assessment of their current and potential distribution in a changing environment. Where studies have been undertaken, they have tended to be molecular or pharmacological in nature [28,29]. However, there is evidence that precipitation change affects species in this family [30], thus, magnifying the need for continuous modeling based on new sets of scenarios in order to fully understand species niche dynamics and ensure sustained NCP.

## 2. Materials and Methods

ArcMap 10.6.1 (ESRI Inc., Redlands, CA, USA) was used to manage collected geographical data, to calculate secondary data, and to visualize maps.

### 2.1. Occurrence Data

Species occurrence data (Figure 1) were collected from the Global Biodiversity Information Facility (GBIF) [31], a database with open access data of where and when species occur, collected from institutions around the world. Selected taxa were from the following 4 genera: Indogofera, *Crotalaria*, *Senegalia*, and *Vachellia*. *Senegalia* and *Vachellia* are among 5 newly recognized genera, consisting of trees and shrubs [32]. The dataset was cleaned to separate historical from current data, with only records from 1960 to present retained for this study. All points outside the country border were omitted. The geographic distance matrix generator [33], which outputs geographic distance between each pair of localities, was used to exclude duplicate records and reduce sampling bias, which leads to overfitting. Using this approach, the risk of deleting other presence points is minimized [34]. As a result of cleaning, occurrence data were significantly reduced but still fit for analysis as they were comparable to previous studies [35–37].

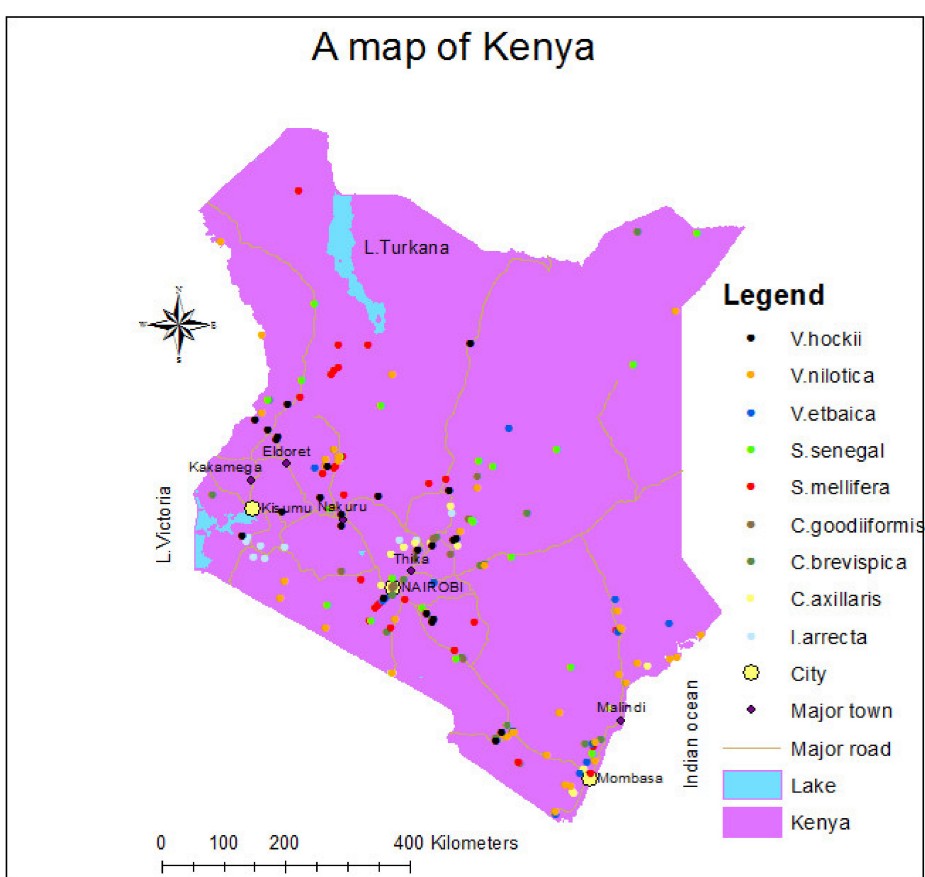

**Figure 1.** Map of Kenya. Adapted from maps downloaded from The IGAD Climate Prediction and Applications Centre (ICPAC) geoportal.

We used nine species in our models (Table 1). The species uses for *Senegalia* and *Vachellia*, shown in the third column in Table 1, are drawn from the field guide by Dharani [24].

**Table 1.** Selected Fabaceae species from the Global Biodiversity Information Facility (GBIF) database, number of cells in which they occurred and their values (extracted from [24]).

| Species Name | No. of 1 km Cells with Species Present | Species Values |
|---|---|---|
| *Indigofera arrecta* | 16 | Medicine, dye, food |
| *Crotalaria axillaris* | 17 | Medicine |
| *Crotalaria goodiiformis* | 16 | Fodder, fuel |
| *Senegalia brevispica* | 24 | Forage, fencing, fuelwood, medicine |
| *Senegalia mellifera* | 31 | medicine, fuelwood, fencing, beehive, soil conservation |
| *Senegalia Senegal* | 23 | Gum, resin, fuelwood, fencing, medicine, fodder, forage, soil nitrogen fixation |
| *Vachellia etbaica* | 16 | Fodder, beehive, medicine, poles, fuelwood |
| *Vachellia hockii* | 24 | Fuelwood, fodder, medicine, construction, soil nitrogen fixation |
| *Vachellia nilotica* | 41 | Fencing, furniture, fuelwood, gum, tannins, forage, medicine, dye, riverbank stabilization |

## 2.2. Environmental Variables

On the basis of previous studies [38–41], this study selected bioclimatic, topographic, soil, and social variables as candidate predictors of species distribution (Supplementary Table S1). Bioclimatic variables represent annual trends, seasonality, and extremes in climate conditions, and hence are important in defining eco-physiological tolerances of species. Soil texture and fertility exert influence on plant establishment while human-induced disturbances play a role in soil management, species selection and dispersion. Elevation is correlated with climate and soil factors, hence may influence species composition. Land use can affect the capacity of a system to support various species.

Historical climate data comprising 19 bioclimatic variables [42] were downloaded from the WorldClim website (www.worldclim.org) at 30 arc second (approximately 1 km) resolution. The WorldClim version 2.1 historical data covers the years from 1970 to 2000 and can be viewed at various resolutions. Warmness index was calculated from monthly mean temperature [43] using the WorldClim data. Digital elevation model (DEM) derived from the Shuttle Radar Topography Mission (SRTM) elevation data was also downloaded from the WorldClim website and used to generate slope and aspect using ArcMap.

Edaphic factors were drawn from soil attributes from the soil map layer of Kenya done by Kenya Soil Survey (KSS) in 1982 and revised in 1997 (ICPAC_IGAD_UNOSAT, https://www.openstreetmap.org/copyright), while distance to rivers was calculated from the river network layer of HydroSHEDS [44] using ArcMap. The high resolution land cover map Sentinel2 prototype (http://2016africalandcover20m.esrin.esa.int/download.php) was also used. In addition, road density was calculated in ArcMap from the major roads of Kenya layer (https://datasets.wri.org/dataset/major-roads-in-kenya), downloaded from the World Resources Institute (WRI) website. The map is an output of the project on Ecosystem Service Mapping.

The human influence index (HII) [45] and population density were obtained from the Socioeconomic Data and Applications Centre (SEDAC; http://sedac.ciesin.columbia.edu). HII is a combination of socioeconomic factors believed to exert pressure on ecosystems such as urban extent, population density, land cover, as well as distance to roads and rivers. It ranges from 0 (close to pristine) to 64 (much degraded areas). HII has been applied in other studies [46,47] and found to be a useful tool for biodiversity assessments under human influences.

## 2.3. Climate and Population Change Scenarios

The RCP gas concentration pathways are used in climate modeling to predict possible future climates up to 2100, where the prediction periods are concatenated as 2021–2040, 2041–2060, 2061–2080, and 2081–2100. Future climate data from MIROC-ES2L climate model [48] mid-century projections (2041–2060) based on RCP 4.5 was used in this study. RCP 4.5 is the second-to-lowest among emission scenarios and could be considered to be a moderate mitigation scenario [49]. In this scenario, forest lands decline in the first half of the century, in part due to expansion of agricultural land to meet

increased population needs, and then expand after 2050 due to shifts to electricity and application of emission prices.

To obtain future bioclimatic variables, projected future climate by the climate model was downscaled and bias-corrected based on WorldClim baseline climate. For downscaling, the projected variable was, first, computed as a relative (for precipitation) or absolute (for temperature) difference between the output of MIROC-ES2L for the baseline and future (2050) years. Relative values are used for precipitation variables to balance out the effect of intense rainfall areas in some regions, which would strain comparison between current conditions and future projections. Differences between current and future climates were interpolated to high resolution grids (1 km), and then calibrated with the WorldClim current data.

Future HII values were computed using potential future population density estimates by SSPs 1, 2, and 3 and corresponding urban expansion. SSPs 1, 2, and 3 represent development that takes a sustainability route, a middle of the road scenario, and a rocky road scenario that is pegged by regional rivalry. In these 3 SSPs, the assumptions on urbanization are as follows: SSP1 estimates up to 92% of world population living in cities and SSP2 estimates 80%, while SSP3 estimates about 60% [50]. While SSPs contain future population density distribution projection, corresponding urban expansion is unavailable. To project future urban expansion, the probability of urban polygon was estimated through logistic regression analysis with population density as an independent variable. Gridded Rural Urban Mapping Project (GRUMP) Urban Extent [51] and Global population density [52] were utilized as responding and explanatory variables, respectively. The regression model used the logit transformation to obtain a linear model of population density $PD$ to ensure that the probability inside urban polygon $P_{\mathrm{urban}}$ would be continuous within the range from 0 to 1 (Equation (1)).

$$P_{\mathrm{urban}} = \frac{1}{\exp(-\beta_0 - \beta_1 PD)} \tag{1}$$

Resultant probability was snapped at 0.5 and all cells above this threshold were considered to be inside urban polygon, while those below were considered to be outside. Then, future HII values for each SSP scenario were calculated by modifying population density and urban polygon scores and by assuming the other component scores of HII were constant. In the total HII score of 64, population density scores range from 0 to 10, while urban polygon scores are either 0 outside the polygon or 10 inside the polygon, as shown in Table 2.

**Table 2.** Scores of population density and urban polygon in human influence index (HII) (based on [45]).

| Population Density (PD) Score | | | | Urban Polygon Score | |
|---|---|---|---|---|---|
| PD (km$^{-2}$) | Score | PD (km$^{-2}$) | Score | | Score |
| 0–0.5 | 0 | 5.6–6.5 | 6 | | |
| 0.6–1.5 | 1 | 6.6–7.5 | 7 | | |
| 1.6–2.5 | 2 | 7.6–8.5 | 8 | Inside urban polygon | 10 |
| 2.6–3.5 | 3 | 8.6–9.5 | 9 | Outside urban polygon | 0 |
| 3.6–4.5 | 4 | >9.5 | 10 | | |
| 4.6–5.5 | 5 | | | | |

## 2.4. Species Distribution and Diversity Indicators by MaxEnt

All layers of candidate predictor variables were resampled to approximately 1 km (30 arc second) resolution in ArcMap. The World Geodetic System 1984 (WGS 84) was used. Then, the layers were stacked in R (R Core Team) to allow simultaneous multilayer analysis. The MaxEntVariableSelection package [53] was used to determine variable importance and output the best predictors. MaxEnt's hinge feature which supports at least 15 samples was used. Using hinge features fits relatively smooth models [54]. During selection, variables were eliminated in a backward stepwise manner ensuring collinearity between explanatory variables was removed. A Pearson's correlation coefficient threshold

of $r \geq 0.85$ and a contribution threshold of 5% were used. The MaxEntVariableSelection uses the Akaike information criteria (AIC), the sample-size adjusted Akaike information criteria (AICc), the Bayesian information criterion (BIC), and the area under receiver operating curve (AUC) to select the best model. Using different information criteria ensures flexibility of choice between model complexity and predictive performance [55]. Beta multipliers between 2 and 6 were tested at 0.5 intervals. Varying of the regulation multiplier has also been applied in other studies [38] to control overfitting.

Then, the results of the variable selection process were used to fit MaxEnt models using the dismo package [56] version 1.1-4 that contains an R link to the MaxEnt model. The current version of MaxEnt (3.4.1) was used in R Studio (RStudio Team). MaxEnt first creates a model, and then uses the predict function in combination with a given set of independent variables to give the probability of occurrence. To get the training and test datasets, we used a 4-fold splitting using the k-fold function, following previous studies that generally adopt between 4 and 10 folds. Therefore, for each model, a 25% sample was withheld for testing. Background area represented space that is currently available and can be colonized or may have been previously colonized by the species being modeled. We used 5000 random background points, similar to previous studies [35,37,57]. The area under the receiving operator curve (AUC) and the continuous Boyce index (CBI) were used to evaluate model performance. The CBI was computed in the ecospat package [58] using default settings. MaxEnt models output the probability of species occurrence; therefore, a certain probability threshold is needed to predict species habitable area. In this study, the threshold at which sum of the sensitivity (true positive rate) and specificity (true negative rate) was highest was adopted as the cut-off for calculating areas of species range. In addition, ranges within species' extent of occurrence were calculated using convex hulls with a single cell buffer.

To assess species abundance and evenness, the Shannon's diversity index ($H'$) [59], which is commonly used was adopted, as well as probability of occurrence of at least one species. Shannon's diversity index measures species diversity in terms of abundance and evenness; the higher the value of the index, the richer and more even a location is in terms of species occurrence. To calculate it, any of natural log, log base 2 or log base 10, can be used. In most cases, the index falls between 1.5 and 3.5 and rarely goes beyond 4.5 [60]. The index was considered to be appropriate, since studies have shown that there is no one ideal index [61]. To calculate it, species occurrence probability output by MaxEnt was first converted into species fraction, and then the index was computed (Equation (2)):

$$P_i = P_i \Big/ \sum_{i=1}^{9} P_i H' = - \sum_{i=1}^{9} \left( P_i \log_e P_i \right) \tag{2}$$

where $P_i$ is the occurrence probability of species $i$ predicted by MaxEnt.

The probability of occurrence of at least one species, $P_{\geq 1}$, was calculated by subtracting the probability of none from one using Equation (3):

$$P_{\geq 1} = 1 - \prod_{i=1}^{9} (1 - P_i) \tag{3}$$

## 3. Results

### 3.1. Climate Change by RCP 4.5 Scenario

Changes in mean annual temperature and annual precipitation between present and 2050, according to the MIROC-ES2L climate change simulation for RCP, 4.5 are shown in Figure 2. Supplementary Figure S1 contains present and future climate distribution in the study area for the selected predictors (Bio1 for mean annual temperature and Bio12 for annual precipitation). In 2050, the mean annual temperature change was projected to be 1.5 °C with the lowest mean jumping from less than −2 °C to more than 5 °C in the study area (Figure 2a), while annual rainfall change varied between −824 and 662 mm with a mean change of −13 mm (Figure 2b) showing a general decrease

in precipitation. Regions that currently receive good rains (Bio12 in Supplementary Figure S1) will experience decreased amounts, while slight increases may be experienced in the southeastern and north coastal areas. During the dry quarter (bio17), the Lake Victoria region will still receive some rains, while precipitation in the coldest quarter (bio19) is likely to increase.

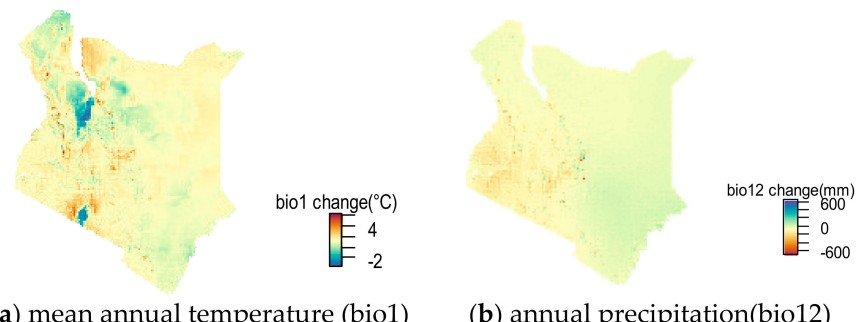

(**a**) mean annual temperature (bio1)    (**b**) annual precipitation(bio12)

**Figure 2.** Projected changes in (**a**) mean annual air temperature and (**b**) annual precipitation between the present and 2050, according to MIRO-ES2L under the RCP 4.5 scenario.

### 3.2. Urban Area and HII Change by SSP Scenarios

According to the projections of SSPs, maximum population density will rise from current levels of just less than 75,000 persons/km$^2$ to about 137,000 persons/km$^2$ in SSP1, 118,000 persons/km$^2$ in SSP2, and 108,000 persons/km$^2$ in SSP3 (Figure 3). Coefficient $\beta_0$ of the logistic model predicting the probability inside urban polygon was calculated as −5.44, while $\beta_1$ was 0.0039, and the regression result for urban projection was statistically significant at *p* < 0.001. Results of urban growth show that the country will experience high urban growth with urban area change from 4686 to 5645 km$^2$, 7379 km$^2$, and 12039 km$^2$ in SSPs 1, 2 and 3, respectively (Figure 4), especially in the central areas and areas around L. Victoria in the western part of the country. Figure 5 shows the corresponding changes in HII values. Areas with HII values greater than 33 will increase from 3878 km$^2$ in current scenario, to 5094 km$^2$, 5465 km$^2$, and 6483 km$^2$ in SSPs 1, 2, and 3, respectively, and the maximum HII will increase from current level of 60 to 62 in all SSPs. Significant changes in HII may be expected in areas between L. Victoria and Nairobi in proportion to the change in population and urban area (Figure 5). The continuum of regions between the lakeside, the capital city, and the coastal area will continue to experience intensive human influence. The coastal area is around the port city of Mombasa and the connection to Nairobi is the highway and railway, and therefore the major trade connection of the country.

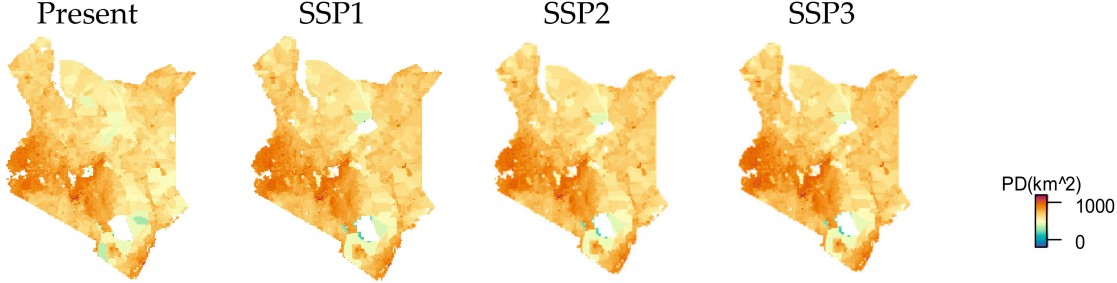

**Figure 3.** Present and projected future population densities according to 3 shared socioeconomic pathway (SSP) scenarios.

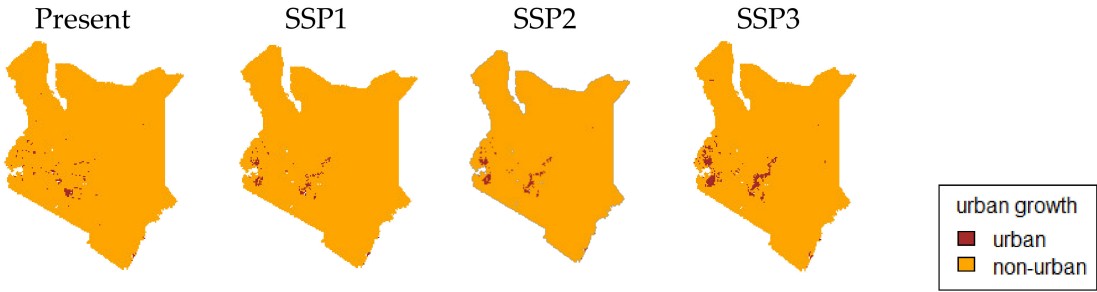

**Figure 4.** Present and projected urban area in 2050 according to population densities by SSPs.

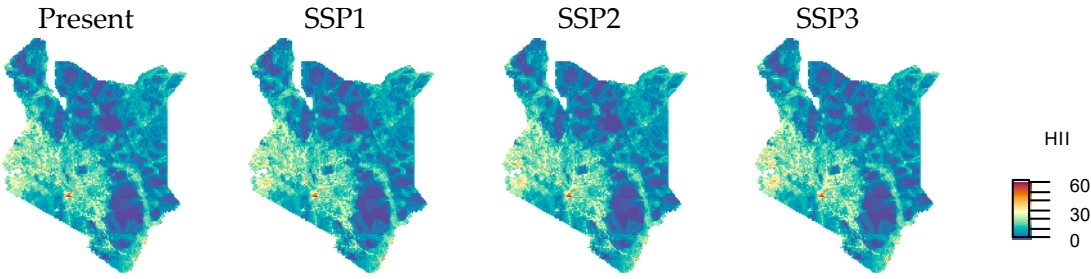

**Figure 5.** Present and projected future HII depending on population and urban area change according to SSP scenarios.

### 3.3. Species Distribution Modeled by MaxEnt

Table 3 shows the predictors for any of the models that were selected from the initial explanatory variables. Beta multipliers were either 2 or 2.5. Predictor maps are attached as Supplementary Figure S1 (bioclimatic) and Supplementary Figure S2 (edaphic/topographic/socioeconomic).

**Table 3.** Model predictors selected by the MaxEntVariableSelection package.

| Variable Name | Description (unit) | Source |
|---|---|---|
| Aspect | Slope direction (8 directions and flat) | Calculated from DEM (Worldclim) |
| Bio2 | Mean diurnal temperature range (mean of monthly mean daily values) (°C) | WorldClim |
| Bio3 | Isothermality (Bio2/Bio7 × 100%) | WorldClim |
| Bio12 | Annual precipitation (mm) | WorldClim |
| Bio13 | Precipitation of the wettest month (mm) | Worldclim |
| Bio14 | Precipitation of driest month (mm) | WorldClim |
| Bio16 | Precipitation of wettest quarter (mm) | WorldClim |
| Bio17 | Precipitation of driest quarter (mm) | WorldClim |
| Bio19 | Precipitation of coldest quarter (mm) | WorldClim |
| CEC | Soil cation exchange capacity ($cmol_c$/kg) | Soil map of Kenya |
| Distrivers | Distance to rivers (km) | Calculated from HydroSHEDS layer |
| Elevation | Height above sea level (m) | WorldClim |
| ExchNa | Soil exchangeable sodium ($cmol_c$/kg) | Soil map of Kenya |
| HII | Human influence index | NASA socioeconomic data and application center (SEDAC) |
| Landform | Features that make up the land surface (14 classes) | Soil map of Kenya |
| PD | Human population density (persons/$km^2$) | NASA gridded population density |
| Rdensity | Road density (km/$km^2$) | Calculated from Kenya road layer |
| Slope | The degree of inclination (decimal degrees) | Calculated from DEM (Worldclim) |

Predictor permutation importance and model test AUC and CBI values are shown in Table 4. For *I. arrecta*, landform had the greatest importance. *C. axillaris* was most sensitive to Bio2 (mean diurnal temperature range), while *C. goodiiformis* was sensitive to Bio19. *S. brevispica*, *S. senegal* and *V. hockii* showed the greatest sensitivity to HII. Both *S. mellifera* and *V. etbaica* showed high sensitivity to distance to rivers. For *V. nilotica*, population density had the highest importance. In models where HII

was selected as a predictor, its permutation importance proved to be high. Landform was also found to significantly affect species distributions. Elevation was important for species in the *Vachellia* genus.

**Table 4.** Percent permutation importance of predictors per model and test area under receiver operating curve (AUC) and continuous Boyce index (CBI) values.

| Predictor | Species | | | | | | | | |
|---|---|---|---|---|---|---|---|---|---|
| | *I. arrec.* | *C. axilla* | *C. goodii.* | *S. brevis.* | *S. mellif.* | *S. seneg.* | *V. etbaic.* | *V. hockii* | *V. nilotic.* |
| Aspect | – | – | – | – | – | 0.0 | – | – | – |
| Bio2 | – | 38.5 | – | – | – | – | – | – | – |
| Bio3 | – | – | – | 13.4 | – | – | – | – | – |
| Bio12 | 0.0 | – | – | – | – | – | – | – | 37.2 |
| Bio13 | 3.6 | 32.4 | – | – | – | – | – | – | – |
| Bio14 | – | – | – | – | 16.3 | – | – | – | – |
| Bio16 | 1.7 | – | – | – | – | – | – | – | – |
| Bio17 | – | – | – | – | – | – | 0 | – | – |
| Bio19 | – | – | 52.2 | – | – | – | – | – | – |
| CEC | – | – | – | – | 17.9 | – | – | – | – |
| Distrivers | – | – | – | – | 41.6 | – | 58.9 | – | 5.5 |
| Elevation | – | – | – | – | – | – | 24.9 | 12.0 | 11.5 |
| ExchNa | – | – | – | – | 3.9 | – | – | – | – |
| HII | – | – | 9.4 | 82.1 | 18.9 | 94.9 | 13.2 | 59.1 | – |
| Landform | 94.7 | 23.2 | 38.4 | – | – | – | – | – | – |
| PD | 0.0 | 0.0 | 0.0 | 1.4 | – | – | 3.0 | 0.0 | 45.8 |
| Rdensity | – | 5.9 | – | – | – | 5.1 | – | 3.1 | – |
| Slope | 0.0 | – | – | 3.1 | 1.4 | – | – | 25.7 | – |
| AUC | 0.94 | 0.95 | 0.93 | 0.82 | 0.83 | 0.94 | 0.77 | 0.82 | 0.95 |
| CBI | 0.85 | 0.45 | 0.85 | 0.83 | 0.70 | 0.90 | 0.80 | 0.65 | 0.78 |

–: NA.

Five of the fitted models had high AUC values of test dataset (>0.9) showing that they had strong discriminative ability to filter suitable species environments from random background points. In principle, AUC values above 0.75 are considered to be appropriate for model discrimination ability; therefore, all nine models were valid for species range estimation. Corresponding ROCs for each model are presented in Supplementary Figure S3. With the exception of *C. axillaris*, CBI values showed averagely well-fit models. Low sampling effort (limitations are discussed in Section 4.5) in highly suitable locations for *C. axillaris* on the southern stretch (Figure 1) may have affected the model accuracy. Therefore, we could not reject the model solely based on incomplete occurrence data.

## 3.4. Species Range Change by Climate and Socioeconomic Scenarios

Table 5 shows the occurrence probability thresholds and calculated ranges for each species according to the climate and socioeconomic scenarios. Ranges within the extent of occurrence are given below those within the potential area of occupancy. Constant ranges across SSPs for *I. arrecta* and *C. axillaris* were because neither population density nor HII contributed to their distribution (see Table 4). Similarly, the ranges for *S. senegal* and *V. hockii* did not change by RCP 4.5 scenario because no bioclimatic predictor was significant in their models.

Under the RCP 4.5 scenario, the total suitable range for *I. arrecta* and *C. axillaris* reduced while that of *C. goodiiformis*, *S. brevispica*, *S. mellifera*, *V. etbaica*, and *V. nilotica* increased. Under a combination of RCP 4.5 and SSP1, no changes were observed for *I. arrecta* and *C. axillaris*. *C. goodiiformis* lost some range, while all other species gained. Under RCP 4.5 and SSP2, *C. goodiiformis* regained lost range and medium gains were experienced by the rest of the species, except *I. arrecta* and *C. axillaris*. Under a combination of RCP 4.5 and SSP3, all species continued to experience gains, except *I. arrecta* and *C. axillaris*. *V. nilotica* experienced huge gains in suitable range under RCP4.5 + SSP1, and under RCP 4.5 + SSP3. This result suggests that projected population density changes are not significantly different between SSP1 and SSP2 as compared with between these two SSPs and SSP3. Species range estimations for the extent of occurrence showed similar patterns, with the exception of *C. goodiiformis* that decreased under a climate change scenario and suffered net loss of range.

**Table 5.** Current and future species ranges at highest sum of sensitivity and specificity. First row represents potential area, while second row represents extent of occurrence area.

| Species | Threshold $P_i$ at Max Sens. + Spec. | Habitat Range (km²) | | | | |
|---|---|---|---|---|---|---|
| | | Present | In 2050 | | | |
| | | | RCP4.5 | RCP4.5 + SSP1 | RCP4.5 + SSP2 | RCP4.5 + SSP3 |
| *I. arrecta* | 0.09 | 117,296 | 111,154 | ← | ← | ← |
| | | 75,117 | 69,620 | | | |
| *C. axillaris* | 0.31 | 37,511 | 21,715 | ← | ← | ← |
| | | 28,218 | 19,923 | | | |
| *C. goodiiformis* | 0.08 | 70,167 | 71,289 | 71,164 | 71,289 | 71,761 |
| | | 39,565 | 38,871 | 38,762 | 38,871 | 39,184 |
| *S. brevispica* | 0.41 | 64,018 | 69,515 | 75,709 | 77,500 | 81,640 |
| | | 63,261 | 69,195 | 75,270 | 77,049 | 81,140 |
| *S. mellifera* | 0.10 | 139,782 | 144,291 | 144,927 | 145,329 | 146,105 |
| | | 138,110 | 142,101 | 142,714 | 143,112 | 143,859 |
| *S. Senegal* | 0.73 | 76,897 | ← | 78,230 | 78,958 | 80,599 |
| | | 72,352 | | 73,686 | 74,409 | 76,007 |
| *V. etbaica* | 0.13 | 239,066 | 239,530 | 244,055 | 245,058 | 246,817 |
| | | 168,143 | 168,248 | 172,276 | 173,105 | 174,780 |
| *V. hockii* | 0.03 | 171,484 | ← | 172,230 | 172,533 | 173,125 |
| | | 159,996 | | 160,450 | 160,686 | 161,005 |
| *V. nilotica* | 0.44 | 115,545 | 119,552 | 145,650 | 149,882 | 161,890 |
| | | 114,172 | 118,234 | 143,817 | 147,847 | 158,471 |

←: Value is same to the left column because the modeled range was not influenced by variables changeable by scenarios.

Figure 6 shows current species ranges and projected changes in future scenarios. Future range projections are given in Supplementary Figure S5. Both *I. arrecta* and *C. axillaris* will likely lose significant range in the western and central parts of the country, while *C. axillaris* might gain a bit on the coastal zone. Under the SSPs, *C. goodiiformis* will lose range but experience gains in the central areas due to increase in HII. Gains in suitable range for *V. nilotica* in the northwestern tip, the southern and central areas, and the northern coastal areas were observed. *S. brevispica* will most likely be degraded in the central areas but gain near the lakes and the ocean.

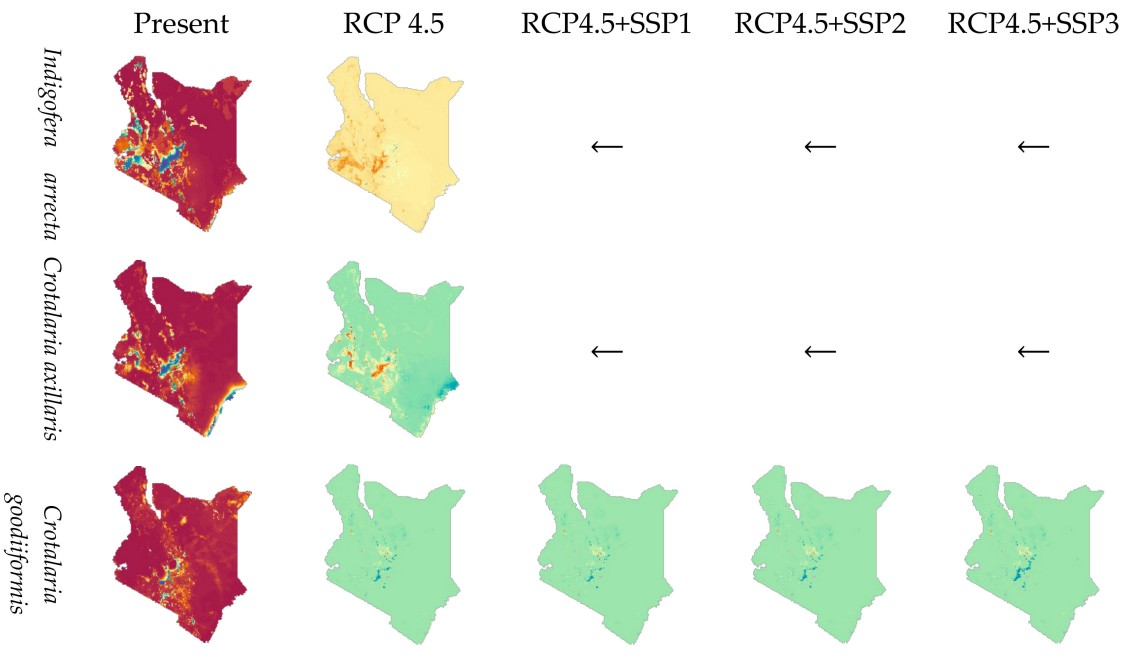

**Figure 6.** *Cont.*

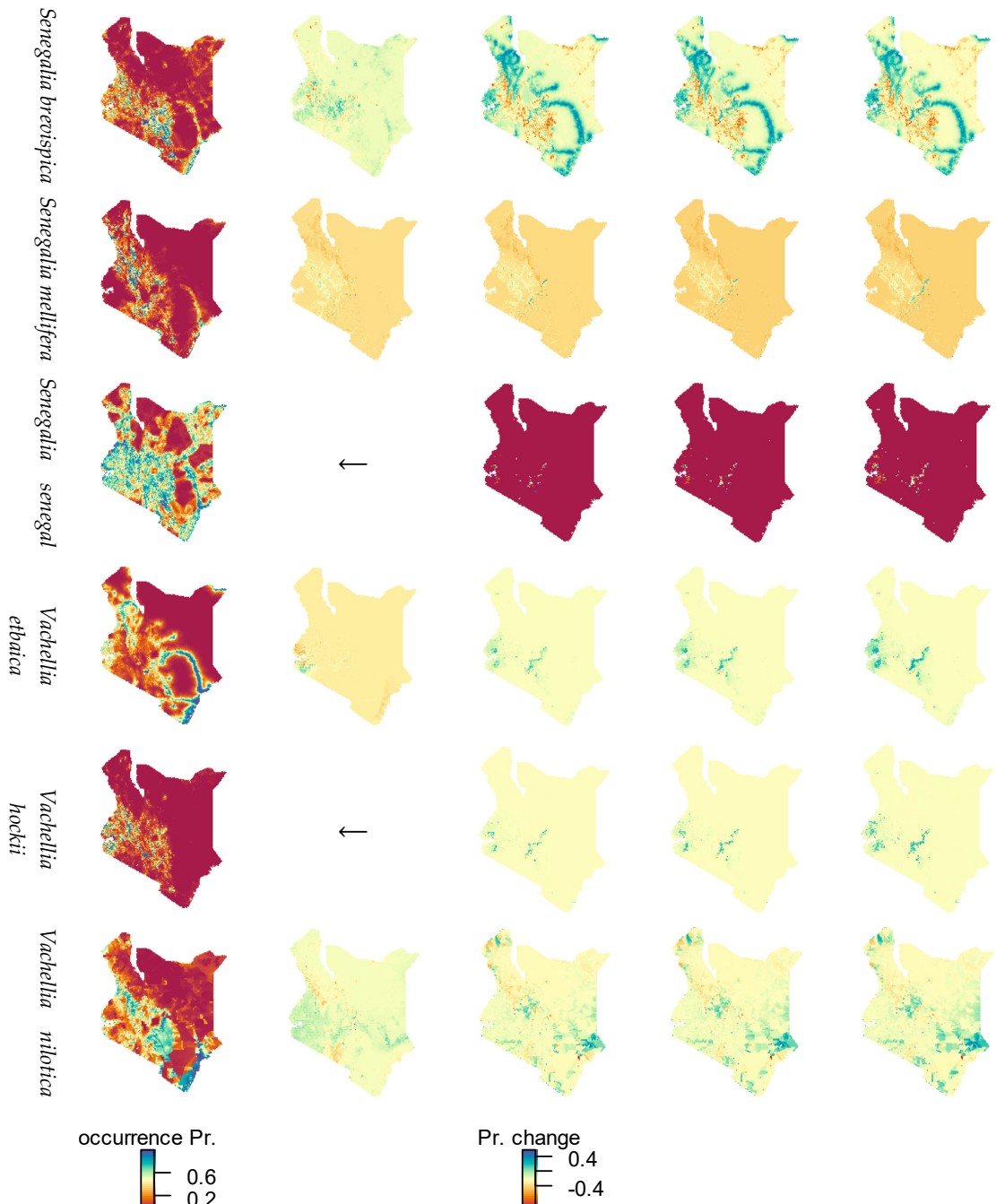

**Figure 6.** Predicted occurrence probability of 9 species by MaxEnt at present and changes in 2050, under RCP 4.5 and 3 SSP scenarios.

Under the RCP4.5 scenario, the Shannon's diversity index showed loss of species abundance and evenness in the western and southern parts of the country. With the introduction of SSP scenarios, losses in western areas were mitigated but more severe losses registered in the east (Figure 7). The increase in species diversity was restricted to the northeastern and central areas. Increases and decreases were higher across all SSPs relative to the RCP 4.5 only scenario. When computed, the mean values of Shannon were highest at present, lower than present but equal across the RCP, SSP1, and SSP2 scenarios, and lowest in the SSP3 scenario.

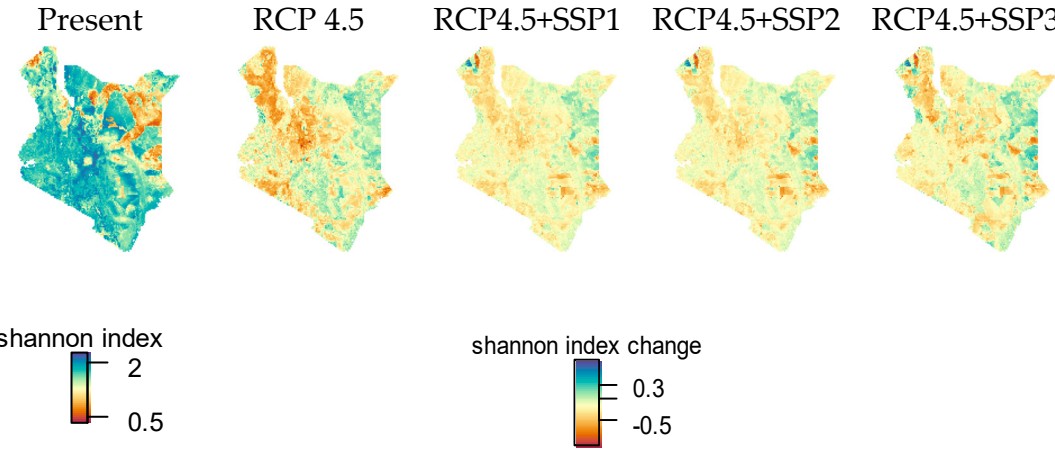

**Figure 7.** Shannon's diversity index at present and changes in 2050, under RCP 4.5 and SSP scenarios.

Figure 8 shows the probability of occurrence of at least one species. There is increased probability of species occurrence across the country under RCP4.5. Under the SSP scenarios, only the eastern parts and northwestern tip experience increases in occurrence probability. Looking at the probability change, there is confidence that the suitability for at least one of the species will increase in the eastern areas. However, some parts that were observed to have increasing probability of occurrence of species were also observed to suffer decreases in the Shannon index (Figure 7). This is due to the excessive increase of only a few species (*C. axillaris*, *S. brevispica*, and *V. nilotica*) in these areas, thus, causing unevenness.

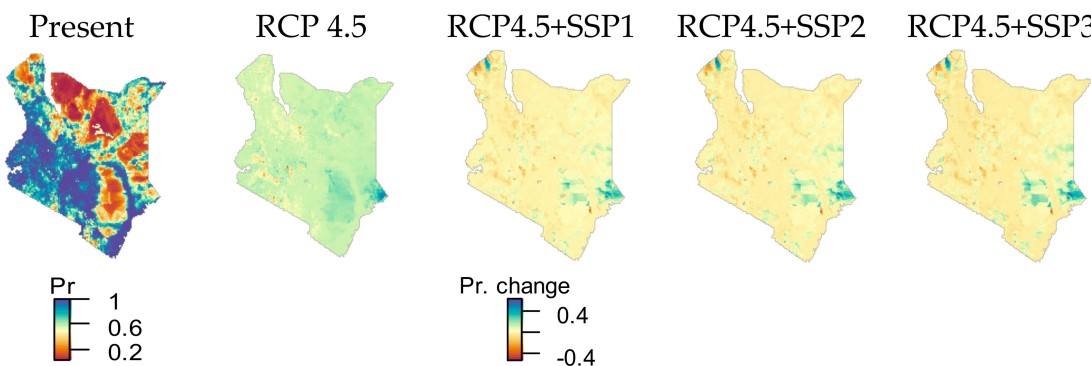

**Figure 8.** Present probability of occurrence of at least one species and changes in 2050, under RCP 4.5 and SSP scenarios.

## 4. Discussion

### 4.1. Species Distribution Modeling Accuracy by MaxEnt

We undertook species ecological niche modeling with some of the rarely used variables such as human influence index, soil characteristics, distance to rivers, and road density. Such variables have been shown to be important for inclusion in MaxEnt modeling [62,63]. The variable selection process resulted in predictors that had high regularization factors such as 2 or 2.5 as opposed to the default MaxEnt regularization multiplier of one. High regularization factors are necessary to reduce overfitting. Test AUC values that were obtained from the MaxEnt models reflect the common values expected for fairly fitted models (Table 4) and were not significantly different from past studies [64]; although [65] asserted that it is very difficult to predict species occurrence with presence only data suggesting a threshold of AUC greater than 0.9. Given this condition, models of *I. arrecta*, *C. axillaris*, *C. goodiiformis*, *S. Senegal*, and *V. nilotica* can be termed as significant. The mapped distributions (Figure 6) of *Senegalia* and *Vachellia* genera are in close agreement with the expected distributions according to the field

guide to Acacias of East Africa [24]. However, *S. mellifera* which was expected to be widespread in the north and northeastern regions did not register much presence there, pointing to a gap in inventory (limitations are discussed in Section 4.5). Distributions of *Indigofera* and *Crotalaria* species are closely similar to those published by the National Museums of Kenya [66].

## 4.2. Environmental Variables Affecting Species Habitats

HII was found to affect the majority of species distributions along with population density (Table 4). Under SSP3 which is projected to have highest population densities, the potential range of species such as *V. nilotica* and *S. mellifera* commonly used by Kenyan communities [28] will likely greatly increase.

For bioclimatic variables, precipitation-related bioclimatic variables were found to affect more species as opposed to temperature (Table 4). One reason could be because most acacias are drought tolerant, and thus less responsive to temperature changes and have been described as dry-indicator plants [67]. Species ranges increased for *C. goodiiformis*, *S. brevispica*, *S. mellifera*, *V. etbaica*, and *V. nilotica* when bioclimatic variables were updated by RCP 4.5 scenario, where the species, except *C. goodiiformis*, belong to the tribus of Acacieae. However, the range for *C.goodiiformis* decreased when only the extent of occurrence was considered. An increase in suitable habitat under future climates for *Vachellia* species has also been reported in other studies [68]. The strong relationship of *C. goodiiformis* to precipitation can be deduced from high permutation importance of Bio19 in Table 4. These findings confirm the assertion that, in the tropics, water availability is one of the important factors determining distribution of plant species. Moisture is especially a driver of spatial distribution of *I. arrecta* whose niche was greatly affected (Table 5) by changes in precipitation of annual (Bio12), wettest month (Bio13), and wettest quarter (Bio16). Its distribution (Figure 6) closely mirrors that of its bioclimatic predictors (Supplementary Figure S1). *S. brevispica* increased in response to temperature-related change.

Although species were mostly sensitive to precipitation changes, a combination of both precipitation and temperature changes greatly affected their distribution. This was illustrated by the significant loss in the suitable range for *C. axillaris* which reduced by almost half in 2050 (Table 5) in response to changes in both Bio2 and Bio13. Decreases in suitable ranges for species that already have small ranges should be seriously considered. There will be a need for more concerted efforts to protect such species under scenarios of precipitation decrease and temperature increase.

## 4.3. Interaction of Climate and Human Influence Changes and Effects on Species Ranges

A combination of climatic and social change further affects species distribution. Across models where either HII or population density was updated, positive trends were registered for six species (Table 5). The suitable range for *C. goodiiformis* first decreased, and then increased marginally across SSPs. The species can currently be found at highly human influenced areas with maximum HII, and therefore little impact across SSPs. This result may be explained by its utilization importance, as it was ranked among the most preferred trees by farmers for reasons such as palatability and drought resistance [25]. This finding shows the importance of including social change in addition to climate change when investigating species distribution. Bucini and Hanan [69], in their study on tree cover in African savannas, concluded that although rainfall is a primary driver, it is not sufficient to explain distribution without accounting for other factors such as human population, grazing, and cultivation. Other studies [70] have shown that human use often overrides other predictors in determining the establishment of species that offer multiple uses to humans.

The spatially distributed Shannon's diversity index (Figure 7) of the studied species showed that the highly diverse areas are also the areas with concentrated population and high potential urbanization. Similar results have been found in past studies which reported a larger number of flora in major settlements in Kenya [71], higher diversity with higher population density in tropical mountains [39], and higher species richness in urban than rural areas [72]. Although the HII index gives little importance to population dynamics (Table 2), human centers in Africa tend to be located in areas of high population growth [73] that are biologically rich with stable climates. Figure 8 shows that

areas classified as the landform of plains (Supplementary Figure S4) and receive low amounts of rainfall (Supplementary Figure S1) are largely less diverse. This result agrees with that of Yusuf et al. [74] who reported that grassland plains tended to experience less human pressure, hence less density of species. In this study the areas with high probability of species range occurrence (Figure 8) coincide with the cropland and urban area land cover categories (Supplementary Figure S4). Agriculturally intensive areas were also observed to have high tree cover by Bucini and Hanan [69]. In general, for the studied species, the results mirror past findings that, in Africa, of the top drivers of biodiversity in savannas and tropical forests, human impacts are most critical followed by climate change impacts [75].

### 4.4. Possible Implications for NCP in Future

These findings show varied species responses to climatic and socioeconomic changes. While some species are favored by climatic change, others are negatively affected. Final suitable ranges for *I. arrecta* and *C. axillaris* were lower than current ranges, meaning that these species will suffer a net loss in range. Such lost range results in communities losing both instrumental values and relational values (Table 1). Relational values may also include indigenous technical knowledge concerning the species and community beliefs and practices relating to the species.

Social change leads to mixed changes in species ranges. For species that are highly exploited, increased human pressure during periods of drought can further lead to their degradation and loss of NCP. However, such species are also continuously being introduced on landscapes due to dispersal by rail, road, rivers, and humans. This trend is likely to continue under SSP3 [76] and may lead to their increased abundance. Moreover, in an urbanizing Africa, where resultant smaller family units are less energy efficient [77], projected demand for both fuel and fodder due to higher meat consumption [15] may lead to deliberate introduction of some species. Such increase in only a few species reduces diversity. This has been demonstrated by the results where areas with increased likelihood of occurrence of species also suffered diminished evenness. Onaindia et al. [78] also observed a similar trend where high human transformation negatively affected species evenness. There is a need for the SSPs to assess not just deforestation concerns but also forest degradation [79], critical for biodiversity conservation and sustaining NCP. Studies on the hotspots of land degradation in Kenya [80] have so far only included the dominant land cover types (grasslands, croplands, forest, and woodlands) while masking urban areas where greatest population change is projected to occur.

Under the RCP 4.5 + SSP3 scenario, where urban growth in Kenya in 2050 will be more expansive than compact, species-specific management will be required, as results have revealed the greatest loss of mean species diversity under this scenario. Management will also be critical in the SSP2 scenario that does not have explicit energy-access policies, and therefore the transition away from traditional biomass takes much longer. The moderate increases in species' potential ranges seen in the results of this study may offer opportunities for alternative exploitation and increased productive activities (thus increasing NCP) but they do not seem to be large enough to account for the foreseen increase in energy demand (thus decreased NCP).

### 4.5. Study Limitations

This study has given a fair analysis of predictors for the studied species but with some limitations. First, spatial biasness of species occurrence data due to low sampling effort in areas with low accessibility [71] could not be eliminated. Presence records are often purposefully sampled from protected areas, hotspots, and locations with easy access such as roads, thus, giving rise to sampling bias. The distribution of *S. senegalia* for instance closely follows roads (Supplementary Figure S2) and overestimates areas where the species were expected to occur especially in the central and western parts of the country near L. Victoria. But potential distribution maps (https://vegetationmap4africa. org/Species/Species_distribution.html) suggest that the species is ubiquitous in Kenya. Spatial bias of collections is a challenge for MaxEnt which assumes that presence data are an independent sample from the species' unknown occurrence distribution [81].

Second, some factors affecting HII values such as road density could not be updated in future scenarios, due to a lack of suitable road models for the future. The contribution of agriculture could also not be modeled due to the complexities of determining irrigated vs. non-irrigated area and how proportions may change in future. Consequently, such factors were assumed to be constant. In addition, factors such as interspecies competition and species adaptability in a changed climate were not analyzed. In general, species distribution models treat species independently of each other.

Third, although this study predicted species range expansion and contraction under future urbanization and HII changes, actual linkage mechanisms between ranges and human influence were unknown. Further investigation of practices that contribute to species range change is required. The potential NCP implications discussed here require comprehensive assessment of net changes by valuing each species' contribution to a range of human wellbeing components.

## 5. Conclusions

This study modeled ecological niches of nine species from four genera in the Fabaceae family using the MaxEnt modeling algorithm. Projections of potential species ranges were then made under current and future climate and socioeconomic scenarios, resulting in 45 potential range maps. Climate impact was assessed by contrasting current distributions with future distributions using projections by MIROC-ES2L and the RCP 4.5 scenario, while socioeconomic trends were analyzed using three SSP scenarios. The study revealed that climatic factors, in combination with human influence, are critical determinants of the distribution of the studied species. Climate change was found to have much more detrimental effects on species ranges as opposed to population change. For the majority of species, population was positively related with species abundance. However, it cannot be assumed that an increase in species diversity will continue indefinitely with increased population growth. In the urban areas, policies for land use are going to be needed to protect the status of species. Some species ranges tended to reduce with change in climate and social conditions. Such reduction potentially leads to loss of NCP, as communities can no longer benefit from instrumental and relational values. Monitoring of such species will have to be introduced in the settled landscapes. The western part of the country should especially be monitored and investigated, since it will undergo drastic changes in both climate and demography. From the maps, it can be concluded that the potential natural range for *Crotalaria axillaris* is quite narrow, thus, calling attention to wise use and development of its suitable habitat areas.

The potential range maps generated should be interpreted with caution; with the unavailability of absence data, the modeled maps represent spatial degree of suitability of the species and not necessarily their realized distribution. The models can further be improved by unbiased sampling that ensures proper documentation of geographical reference data and event dates. Some species data were last recorded more than 15 years ago reflecting the need for an updated database. This will be the next step in the expected continued trend analysis of species distribution. In addition, this study can be expanded to include the full set of RCP-based climate change scenarios.

**Supplementary Materials:** The following are available online at http://www.mdpi.com/2225-1154/8/10/109/s1, Figure S1: Bioclimatic predictors selected by the MaxEntVariableSelection package for any species in present and in 2050 by RCP 4.5., Figure S2: Soil, topographic, and geographic predictors selected by the MaxEntVariableSelection package for models, Figure S3: MaxEnt ROCs for the 9 modeled species, Figure S4: Landcover map of Kenya, Figure S5: Projected occurrence probability of 9 species by MaxEnt in 2050, under RCP 4.5 and 3 SSP scenarios., Table S1: Initial environmental variables.

**Author Contributions:** Conceptualization, R.N. and T.M.; methodology, R.N. and T.M.; software, R.N. and T.M.; formal analysis, R.N.; investigation, R.N. and T.M.; resources, R.N. and T.M.; data curation, R.N.; writing—original draft preparation, R.N.; writing—review and editing, R.N. and T.M.; visualization, R.N. and T.M.; supervision, T.M; project administration, T.M. All authors have read and agreed to the published version of the manuscript.

**Funding:** This research received no external funding.

**Acknowledgments:** The authors wish to thank Chihiro Haga for his technical support. We also thank the anonymous reviewers for their comments to prior versions of the manuscript.

**Conflicts of Interest:** The authors declare no conflict of interest.

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
