# Peer review of "Potential Effects of Climate and Human Influence Changes on Range and Diversity of Nine Fabaceae Species and Implications for Nature’s Contribution to People in Kenya"

_climate, doi:10.3390/cli8100109_

Round 1

Reviewer 1 Report

I have read the authors' responses to my comments (I am Reviewer #1) and I enjoin that they have done an excellent job of addressing my previous concerns and suggestions in this revision.  As I mentioned earlier, this paper represents a nice proof-of-concept articulating exemplary methods to integrate climate change and human land use decisions to predict changes in species' distributions and diversity using plant species of value to people. 

Author Response

Dear Reviewer,

We thank you once again for your review and comments, which we used to improve our manuscript. We are happy that you consider our work to have been satisfactorily done.

Reviewer 2 Report

Overall this manuscript has some interesting ideas, but it was difficult to decipher a great deal of it. Aside from a lot of smaller points, I really struggled to understand the main research questions, what results are simply output of existing models, and what here is both new and needed.

Species Data. One large concern I have with the manuscript is that it seems like there are very few actual data points of the nine species across the nation. As a result it seems like the underlying data may be too sparse to actually build any large extent models as have been done here.

Introduction. The Introduction was very scattered and disorganized in terms of providing a solid framework for what research gaps you are seeking to address in the manuscript. I found much of it to be unrelated to the objective of the research. My suggestion would be to start over with a new Introduction that you aim to have 3-4 paragraphs total. You don’t need to describe everything about these topics, but have rather start with a general Introductory paragraph that relates to the topic you are interested in (e.g., how climate change will influence forest dynamics). Then in the next 1-2 paragraphs provide some indication of what we know about this topic and then where the gaps are. Finally, your last paragraph should have an overarching goal or aim of the research and then specific questions/hypotheses/objectives that seek to address that overarching goal/aim. When I read the Introduction I really did not see why acacias are important to study, what we need to know, what is already known about them, etc. Note, paragraphs should contain a minimum of three paragraphs.

L98. Please explain this database more. What does it contain? From what time periods? How accurate are spatial locations?

L101. I would not consider 1960 as current. Also, I’m really unclear what data you are trying to access and what you want to use it for as the Introduction was very unclear.

L103. I don’t understand what you are doing with the software and what is being cleaned.

L106. How were the comparable?

L107. Work to integrate table and figure notation in parentheses. Thus, here you can say “We use nine species in our models (Table 1).”

L108. Why 1 km?

L122. Cut this first sentence and in previous paragraph just put in parentheses the table notation. The table should be in main text, not supplement. The table should describe each parameter and give a citation for why it is important to consider in your work.

L122. Incorrect place for ArcGIS notation. Need to provide in paragraph when discussing the geographic data.

L148. Are these from ICCP models? If not, explain why. I was not very clear on what types of models you are using to guide your work here.

L164. I really didn’t understand these projections or model assumptions. I would suggest adding a table that lists all of these different scenarios and describes them.

L172-181. I don’t understand what is being done here. What are the input data describing, what is this analysis doing, and why?

L185. Need to indicate the datum, project, and pixel size/minimum mapping unit of all input data before describing this rescaling. Also, how many data layers are being coarsened?

L186. Need to provide more information about what Maxent is doing and describe the input variables. Ultimately I’m not clear why this software is being used.

L192. Maxent uses these or just provides them and you use to decide?

L195-97. I’m not sure what this is referring to.

L198. Need to indicate which variables were removed.

L208. Why are species diversity metrics being measured? This doesn’t fit in with what I thought the objectives of the research were.

L221. Isn’t this entire subsection just known results? That is, aren’t these types of models already produced for the area?

L236. Again, how much of this model output is simply presenting work that is published elsewhere? If these are standard scenarios, what’s new here?

L277-325. In my mind this was the part of the manuscript that was new. But I didn’t really understand fully what you are presenting in a clear and concise manner.

L327. Discussion should begin with answers to your main research question. Also, please avoid all of the subheaders in the Discussion section. Finally, the Discussion is overly long and has aspects that are unrelated to the main research project.

Author Response

Thank you very much for your review.

Once again, thank you very much.

Reviewer 3 Report

The manuscript entitled with “Potential Effects of Climate and Human Influence Changes on Range and Diversity of 9 Species and Implications for Nature’s Contribution to People in Kenya” focusses on species distribution modelling under current climate and projections into future climate scenarios of 9 plant species. The methodology has still room for improvements. Please see my detailed comments and recommendations below.

Specific comments:

L3: You would usually write out numbers form one to twelve; I think you should add “plant” species or the family name in the title

L62: In this study, …

L63f: The sentence starting with “Fabaceae is one..” could be added to previous one for main message

L81: the paragraph on maxent modelling breaks the reading flow. Consider moving this part up; The paragraph starting in line 61 would usually be the last for the introduction

L94: delete this line, it is not needed

Figure 1: I don’t think this figure is needed here as it does not bring much information. If you want to keep it, consider adding the occurrence points or any other useful information

L99 and elsewhere: latin names of families/genera/species should be in italics

L107: “the second column…” no need for this sentence. It becomes clear form the table

L110: that GBIF is the short form should be indicated after Global Biodiv… in line 98

L111: consider using “usage” instead of “values”; values is like utilitarian values but here it rather means use or usage by people or function in the ecosystem when it comes to nitrogen fixation

L115 “distributions”; you could drop several time the word “variables” retaining only the last

L116 “which are” or “and hence” instead of “hence”

L125 “comprising” instead of “consisting”

L135ff: please reword this sentence; you included the land cover map is the main information here, not the ESA initiative

L186: please also cite R here (and RStudio if you used it)

L201ff: please reword this sentence like “We used a 4-fold splitting …” or else

L214f: did you convert into presence/absence of a species? Did you use a threshold? It does not become clear to me here. Consider rewording.

Figure 2: the maps could be larger while the legends could be slightly smaller

L249 “between Lake Victoria and Nairobi…” is better; the coastal area is around Mombasa and the connection to Nairobi is the highway (including railway) and therefore the major trade connection of the country. Might be worth adding here or in the discussion.

L256f: no need to name the R package here again. I was wondering in the methods section, did you not consider beta multipliers <2? For all models they are pretty low here. I usually use 0.5-4.0 in 0.5 increments and never get values higher than 3 as preferred models (I use ENMeval package by Muscarella et al. 2014)

L270ff: you are right, AUC tells how well the model discriminates between presences and the random background. Warren et al. (2019, https://doi.org/10.1111/jbi.13705) and others however detailed that AUC does not mean anything for maxent models. You should consider also giving the continuous Boyce index here (as easily calculated in ecospat package) or omission rates under minimum training presence or testing of null models like Bohl et al 2019 or Raes & ter Steege 2007 introduced (https://doi.org/10.1111/jbi.13573; https://www.jstor.org/stable/30244521). Relying only on AUC is not appropriate for maxent models

L278: “MaxEnt models output..”  instead of “MaxEnt model outputs”

L279: here you introduce the threshold I was asking for in the methods. This should be given in the methods section and not here

Figure 6: for some species, the SDM (and also the projection) look like showing artefacts from sampling like occurrences along the major roads (e.g., V. etbaica). From my point it seems unlikely that this pattern represents the actual distribution

L328: consider using active voice (here and elsewhere). E.g., “We used ecological niche modelling with….”

L333ff: As I mentioned before, AUC values are not ideal for maxent model evaluation. If you are confident about your models (as I saw from response to reviewers comments from the first round of revision), consider including other evaluation metrics like CBI, orMTP or null models. This would massively increase reliability of your models. Another issue you mentioned here is the mismatch between model output and distribution maps from the field guide you cited. This is of course due to sampling bias (e.g., along roads as possibly illustrated in fig. 6). This could be tackled by using clamping (e.g., in ENMtools R package) or reducing the background area, so that the model can produce broader predictions.

L372: is this also true for V. etbaica which was mainly predicted to occur along the major roads? Or is it rather sampling bias? This should also be discussed here

L392: here is a reference issue

L426: now you give many of the explanations I was asking for in previous comments. I would recommend to include early in the discussion, in the first case when you hit a point for this kind of explanations, you include a short note in brackets like: “(limitations are discussed in section 4.5)” or so

L447: “using the MaxEnt modelling algorithm”

L454 “For the majority of species…”

L457f: please reword; a bit unclear

Reviewer 4 Report

I found this study sound and interesting. I have detected no major issue that would call this work into question. Hence, I endorse its publication. Note, however, that I am qualified mostly as an ecological modeler and have only partial knowledge in plant biology.

I would offer three comments:

1. Did the authors validate in any way the current species distribution output by Maxent? I feel that this is quite important in the frame of a large, long-term modelling exercise, as it serves as the base for predictions. Furthermore, Maxent output cannot be blindly trusted (see next point as for the reasons).

2. As said, I think this work meets best practice and state-of-the-art quality. However, there is an inherent limitation to all modelling exercises relying on Maxent, namely that the latter is purely correlational by nature. The modelling results are therefore not based on mechanistic simulations. In the present work, this translates into unreliability of the distribution maps (both present, as discussed in the previous point, and future, as we discuss here). Indeed, Maxent would for instance predict presence of a species as soon as the environmental conditions (i.e. explanatory variables thresholds) are met, and totally disregards the processes that physically allow for colonization, since mechanistic processes such as dispersal are not reproduced. Hence my criticism and question to the authors: are they confident that dispersal of the species considered can occur from their current range to the regions in which they are predicted? 

3. Minor, but there are a few typos and awkward sentences here and there, so I would recommend that the authors check again thoroughly their manuscript before resubmission. (Really nothing big, as the manuscript is of very good quality altogether.)

Best wishes,

Round 2

Reviewer 2 Report

Overall I was disappointed by the lack of revisions and serious consideration of my previous comments directed at improving the manuscript. The Introduction required an entire rewrite, not a single cutting of a paragraph and adding of another. Also, the manuscript remains very disjunct throughout and is making a much larger story than it really is about. In the end, this work needs to really be rewritten as a simple analysis of evaluating niche modeling based on future climate for these species. It isn't about anything other than that and hence should not be a long paper, or trying to integrate other perspective disciplines.

Reviewer 3 Report

You did a good job including my suggestions and the manuscript looks good to me now. One minor thing that could be changed: the background color of Kenya in Fig. 1. Would you mind printing it in light gray or so?

No further comments from my side.

This manuscript is a resubmission of an earlier submission. The following is a list of the peer review reports and author responses from that submission.

Round 1

Reviewer 1 Report

review of "The Cross-Effect of Climate and Human Influence Changes on Potential Distribution and Diversity of Useful Fabaceae Species in Kenya"

The authors modeled changes in geographic range and species diversity for 9 species of woody plants in Kenya due to the interaction or regional climate change and several scenarios of socio-economic human development using the modeling package MaxEnt.  These plants were selected for this study because these species are of economic, public health, and likely also cultural importance to people in Kenya, and in addition, reasonably decent occurrence data are available on these species for input into MaxEnt.  In my opinion this paper contributes nicely to growing discussions about how to integrate ecological data (such as land use and climate change) with data that are harder to measure representing social and cultural capital for the species being modeled.  This paper represents a nice-proof-of-concept articulating exemplary methods of this integration.  In addition, the results revealed interesting and complex interactions of taxa with both drivers (climate change and human influences) – no simple response was found among all study species.

I have a number of technical comments, see below, but my general suggestions would be to strengthen tan important underlying message of the study.  The "nature's contributions to people" (NCP) concept lays out a deeper direction for natural resources valuation than is typically embedded in the construct of ecosystem services.  This was an important message in the IPBES 2018, UNESCO 2018, Diaz etal 2018, and others.  Bring more of this discussion into your opening paragraph – this is not just nuance – this is an interesting and important point that calls for rejection of the prominence of purely utilitarian modeling in natural resources valuation.  Modeling can incorporate predictor dimensions from social and cultural capital and these models need to inform the development and implementation of environmental, social, and natural resources policy for sustainable development.  I suggest returning to this and strengthening the last part of the discussion, as well.

specific comments by line number:

lines 2-4_  "Interaction of.." instead of "Cross-Effect of…"  Title needs revision for clarity.  The term "interaction" works better – in fact this term is used widely in the IPBES 2018, UNESCO 2018, and other documents discussing this topic.

line 11_   why "useful" ? 

lines 12-14:  dense jargon for anyone not familiar with MaxEnt – simpler summary at this point?  The abstract is an important device to hook readership

line 15:  for clarity, use "ranges under these scenarios were modeled using…" and delete "with scenarios" after the MaxEnt

line 16: Shannon entropy…  very dense, Shannon Diversity Index is widely recognized (see more on this below)

line 24: define "NCP" here

line 20: why did these ranges expand?  (see below)

line 22-24:  Strengthen the take home message:  critical management is important to prevent biodiversity loss, ecosystem dysfunction, natural resources loss, NCP loss, etc., (see comments above on NCP)

line 30:  "bequeathed to man"   please reword "bequeathed to man".  Please avoid this kind of gender pronoun use, and elevate the discussion about valuation – do you mean ecosystem services as utilitarian valuation? cultural, aesthetic? etc.  Typical discussions are restricted to only the former – see my general comments above

line 39:  include ."both climate change and human urbanization land use change

line 40:  that report uses the word interaction on multiple occasions, not "cross-effect" (see comment above)

line 46-49:  move down to new paragraph, after presenting all of the general background information on Kenya in lines 50-56, and expand on the goal of these activities – it is more than just to determine where changes may occur – scenarios and models inform policy development as is stated but only in passing (and consider dropping in a few of your preferred citations on that).

line 58: typo, missing bracket

line 61:  consider adding "…for thousands of years."

line 66: "adversely affect" how?  --  just a phrase or two, foreshadow your own results

line 66-68:  that's the sentence I asked for just above

line 70:  need citation(s) for "ecological niche modeling"  Franklin 2009 set the standard and Elith etal 2010 has an excellent summary of species distribution modeling and MaxEnt in particular using exemplary case studies.  Elith etal 2011 (your citation #42) is a more of technical treatment.

line 77: (and elsewhere, e.g., line 83, ):  standardize the spelling using "MaxEnt"

line 91: delete first comma immediately after "SSPs"

lines 86-101:  this is a mixture of introductory section explanation of the SSPs concept, but then it also included detailed loadings for your models that really belong in the Materials & Methods section.

line 103:  what is meant by "useful"?  Refer to the opening parag NCP concept in your justification.

line 103 and Table 1:  Why did you specifically select these 9 species?  solely based on data availability?  wide geographic ranges of locality data? These seem to have a wide range of "use values" – was that part of your selection process, too?

line 132, Section 2.2:  This is a nicely constructed and well explained section --  very clear and reproducible.

line 138:  insert comma after "soil factors"

line 132, Section 2.2 & 2.3:  These are nicely constructed and well explained sections --  very clear and reproducible.

line 197, Section 2.4:   This is a much denser section and carries a number of important judgement calls, all of which are perfectly defensible.  My comment is to drop in a few more citations to emphasize that this paper is performing very standard methods using MaxEnt.  In addition, better documentation will broaden the use of this interesting approach.

line 225:   Typically Shannon's diversity index uses the natural logarithm, ln() or loge , not log2 as in indicated in the equation.  Which did you use?  If log2, then explain why, please.  One other suggestion I would make on using the Shannon would be that if you use loge(), then you can calculate the exp(H'), and that gives you what can be called "an equivalent number of equally common species".  Try it out for pi's of (0.6,0.1,0.1,0.1,0.1) vs (0.2, 0.2, 0.2, 0.2, 0.2).  The point is that exp(H') has units of species number and is more readily interpretable in qualitative arguments.  Of historical interest, you might be interested in learning that Robert MacArthur in the 1970's was the first person I know of to use the exp(H') with the wording in quotes above, but his use was not widely adopted.

lines 281 to 290:  the text in this paragraph does not match with all of the entries in Table 5.  Please recheck the descriptions.  In addition, it is redundant with Table 4 (you might consider listing the predictors in Table 4 by row in descending order of their loading).  Perhaps a better way to present this might be to say which species show high predictor contributions for each of the major rows of table 5.  For example, I note that only V. nilotica shows a high coefficient for population density.  Is this because it is common in rural areas as an important range fodder ?

general comment on the first part of the Discussion:  how might ranges expand or contract?  What roles are played by people under these SSP's?  Do people plant these species or land use change removing them? seed dispersal by livestock? species ranges will be connected positively and/or negatively by direct agricultural practice and indirect agroecosystem function.  What role does cultivation play in the model since these are "useful" plant species? Can any claims be made about the net increases or decreases in NCP with changes in distributions for these 9 useful taxa?

line 415: "importance of including social change in addition to climate change when investigating species distribution changes" important take home message – expand on this in the intro and discussion. 

line 456-458:  nice prediction, I would tend to agree

Figure 2 (a) legend color coding ranges are blank, (b) is fine.

Figure 3: legend color coding ranges are blank

Figure 5: legend color coding ranges are blank

Figure 6:  simulation result for row (Crotalaria axillaris) and column (RCP4.5+SSP1) is a black and white image in my copy

Figure 6: legend color coding ranges are blank

Figure 7(a) and 7(b): legend color coding ranges are blank

Figure 8(a) and 8(b): legend color coding ranges are blank

Reviewer 2 Report

The manuscript present a sere of models performed for 9 useful Fabaceae species from Kenya. The topic is interesting and suitable for the scope of the journal. However, in my opinion there is an important flaw in the species dataset that compromise all the results. The occurrences for species modelled range from 16 to 41.  From such low dataset authors project the results to the whole extent of Kenya (580.357 km2 in extent¡¡¡). These poor datasets usually produce skewed results that do not reflect the real ecological niche of a given species.  All the results and consequently the discussion are seriously compromised.

I recommend to use complementary data sources (i.e. vegetation maps) to complete the occurrences, and then repeat the analyses. For sure, results are going to change drastically.

Also, despite English is not my mother tongue, I have found many mistakes and many sentences that need to be revise. I have pointed some of them in my report but at some point I have stopped because, in my opinion manuscript clearly needs a thoroughly linguistic revision.

Other general comments

Across the text the maps generated for the different scenarios are interpreted as “map of species richness”, “species range”, “range contraction”, "species maps", etc. However, as they are different scenarios they must be taken as potential outcomes. These type of models only can be taken as niche modelling, so really we do not know if the species range is going to change is such way, we only could assure that species niche is going to migrate, contract, dwindle or disappear. We cannot talk for example of  “range contraction” but of “niche contraction”. Please, put in context this idea across the manuscript.

Specific comments

Title does not reflect the content properly. It seems that all the useful Fabaceae of Kenya have been assessed. Also, this is not properly reflected in keywords, that is another option.

Abstract

Lines 15-16: Revise the sentence in the present way is somehow confusing. My proposal: “Species ranges were modelled using MaxEnt under different scenarios. Results were used to map species richness”

Line 22-23: The sentence is confusing. What is “critical management of biodiversity”? the concept is sounds weird to me. Also, What do you mean with overspecialization and overexploiting? in what sense? Please revise.

Lines 29-32: The sentence is too complex, please rephrase.

Line 42: I suppose not only urbanization but also land use changes and intensification in general.

Line 58: there is a missing bracket in the references

Lines 60-61: “by local people” is missing at the end of the sentence. It is better to place this before product listing.

Line 61: Change “Fabacea” into “Fabaceae”

Line 62: Do you really mean “actual” or “current”?

Lines 62-63: “changed climatic and land cover environment” sounds weird to me. It can be replaced by just “changing environment”.

Line 64: This “unknown reasons” sounds cryptic, What are they?

Lines 66-68: This sentence is very complex, please rephrase.

Lines 71-72: Revise the sentence “Many species distribution models exist ranging from profile to regression models to machine learning methods” in the present form is very difficult to understand. Do you mean that “most of the existing distribution models rely on different methods, from regression models to machine learning methods”?

Line 72: Change “Of these” into “Among them,”

Line 76: Also absence data can depend on uncontrolled factors, such as past  human impacts.

Lines 82-83: Use “Oxford comma (serial comma)” for lists: “Using maxent, bioclimatic, and socio-economic predictors”

Line 83: remove “Using”.

Line 88: Chang “between” into “among”

Lines 90-95: This sentence is very complex and there is a mix of verb tenses in it.

Line 102: it is better “potential future distribution” see my comment on top.

Line 103: “…of selected 9 useful species in the Fabaceae family by possible climate and human influence changes in Kenya”, awkward wording please rephrase.

Lines 104-108:  Please remove these lines. They seem to be misplaced. While some of the ideas belong to material and Methods section, others are more suitable for results.

Material and methods

In my opinion there is an important flaw in the species dataset that compromise all the results. The occurrences for species modelled range from 16 to 41.  From such low dataset authors project the results to the whole extent of Kenya (580.357 km2 in extent¡¡¡). Why do you use this poor dataset for these species, assuming most of them are quite common? I cannot realize how this dataset can model the distribution for the whole country. For example, for Indigofera arrecta you have just 16 points ranging from 200 to 2700 m asl, that contituts a very poor dataset. In other cases such as Crotalaria axilaris, it occupies rivers, swamps, coasts. For this species, I do not know how do you model the species niche since it does not depend on macroclimate data but in microclimatic wetter conditions.

These poor datasets usually produce skewed results that do not reflect the real ecological niche of a given species. In my opinion this situation is only acceptable for narrow endemic species.  

Line 114: Change “was” into “were”

I do not mark linguistic mistakes and awkward wording hereafter. The manuscript clearly needs a thoroughly linguistic review and spell check.

Line 177: “variables” (in plural)?

Lines 182-183: “score” appears three times in the sentence.

lines 111-113: Most of the information is superfluous since details can be checked in the map.

Line 114: Why do you select this species? you need to state the base of species selection here.

Line 120: “exogenous points” sounds weird. I assume you just removed the points that exceed the country boundaries.

Table 3 is more suitable for supplementary material.

Figure 6: Why the central figure in second line appears squared in black?

Line 360: Why is surprising for you that HII proved to be important? According the introduction section this is one of your premises.

Line 481: For me, this is a conceptual error. Distribution models are just projections of the modelled niches for different scenarios. The cannot be assumed as “species distribution maps”, see my general comments.

Reviewer 3 Report

Thank you for the opportunity to review this manuscript.

General comments:
- Background information is missing in the Abstract - it jumps directly in what the study aimed to achieve without any justification of current knowledge and reasoning for the study

- I am concerned with the small sample sizes of presence points (<50 for each of the 9 species; for many species <20), especially given the large number of predictor variables (supplementary Table S1). For example, for Indigofera arrecta, the selected model had 6 variables (Table 4), even though only 16 occurrence points were included in the model (Table 1). I do not think one can reliably model species distribution with such small sample sizes. I recommend restricting the study to those species with at least 20 presence points. I also recommend substantially reducing the number of predictor variables. This would help make the results more credible and also help simplify the paper, as at the moment the manuscript contains a lot of information and is overly long

- The number of references is too large (121 references) and should be decreased substantially

- Throughout the manuscript: use either "MaxEnt" or "Maxent" - pick one and maintain consistency

- Throughout the manuscript: replace "useful species" with "species utilized by humans"

Specific comments:
L11-14: the information is misleading: first the authors state that they analyzed 5 scenarios for each species, whereas next they mention 1 and then 3 pathways. Part of the confusion likely arises from use of jargon that is unfamiliar to many readers. More information is needed to differentiate scenarios from pathways

L16-18: replace "parts of the country that are likely to change in species abundance" with "parts of the country where species abundances are likely to change"

L21-22: it is unclear if the species analyzed are cultivated or grow on their own

L23-24: unclear what "overspecialization" of species refers to. Phrasing needs to be improved - for example, "while overexploiting others" seems a bit disconnected from the rest of the sentence

L24: what is "NCP"? This is the first mentioning of this acronym

L25-26: order keywords alphabetically

L29: insert reference to literature for the first sentence. Also, this sentence has 2 subjects whereas "focus" is singular..

L57: abrupt transition from previous statements. Linkage sentences are needed

L58: close square bracket for reference

L69-70: definition is somewhat vague. "Presence data" of what?

L70: "distribution" of what? Please specify

L111: replace "are" with "is"

L114: replace "was" with "were"

Table 1: column "Use value" - standardize by choosing to report using lower or upper caps only; with or without commas only

L138: did you investigate correlations between predictor variables, excluding highly correlated variables from the same model structure?

L177: explain why relative vs. absolute differences were considered for precipitation vs. temperature

L179: specify grid cell size

L202-203: you mention that variables were removed in a stepwise manner. Did you use forward or backward stepwise variable selection? I am concerned with the small number of observations for some of the species (column "No. of occurrence points" in Table 1), especially given the large number of predictor variables considered (supplementary Table S1). An often-applied rule-of-thumb is that 50 confirmed presence points are used. Yet, none of the 9 species have >=50 presence points, and many have <20 presence points

Figs. 2, 3, 5, 6, 7: legends are missing the gradient color scale

Fig. 6: maps look very similar comparing present and future scenarios. This is not surprising given the large similarities in results listed in Table 6. I recommend eliminating Fig. 6 because the manuscript in its current form has too many figures and tables

Fig. 8 and associated text do not contribute much to the paper and I suggest eliminating. There are too many tables and figures in the current version

Round 2

Reviewer 2 Report

The authors have made a sere of changes that improve the manuscript. However, the main problem, that is the small datasets of occurrences used to model common species and this problem, still persists in the current version. In my opinion, this small datasets for these species projected for an the entire Kenya country cannot reflect the ecological niche of common species like these and I cannot find a justification for this poor dataset.

My advice is to make a reasonable occurrence sampling (field sampling is a good idea) and repeat the analysis with a robust dataset of occurrences.

Reviewer 3 Report

The authors have addressed my questions and comments. The manuscript is improved, but a few further edits are needed.

I suggest changing the title to "Potential Effects of Climate and Human Influence on the Distribution and Diversity of Fabaceae Species in Kenya".

In the Abstract, replace "Species values may explain this positive trend" with "The value that these species hold to people may explain the projected increase in distribution".

The last sentence of the Abstract is a bit counterintuitive - your models show that most species ranges will expand, so do you imply here that expansion of these Fabaceae will lead to degradation of the plant communities/decline of other, "wild" plant species that you did not model? Improved clarity is needed. You also need to mention in the Abstract something about the species that will decline in range (I. arrecta, C. axillaris) and why this might occur.

Fig. 1: delete "Country_boundary" from figure legend. You are showing in blue shading the entire extent of the country, not just the boundary. Also, I recommend replacing the blue country shading with another color. This is because it is difficult to tease apart water bodies from the country

Fig. 8: something seems to have gone wrong with this figure. Why are the scenarios names listed both above and below the figure panels?